# Assessing the Heat Generation and Self-Heating Mechanism of Superparamagnetic Fe_3_O_4_ Nanoparticles for Magnetic Hyperthermia Application: The Effects of Concentration, Frequency, and Magnetic Field

**DOI:** 10.3390/nano13030453

**Published:** 2023-01-22

**Authors:** O. M. Lemine, Saja Algessair, Nawal Madkhali, Basma Al-Najar, Kheireddine El-Boubbou

**Affiliations:** 1Department of Physics, College of Science, Imam Mohammad Ibn Saud Islamic University (IMISU), Riyadh 11623, Saudi Arabia; 2Department of Physics, College of Science, University of Bahrain, Sakhir 32038, Bahrain; 3Department of Chemistry, College of Science, University of Bahrain, Sakhir 32038, Bahrain; 4King Abdullah International Medical Research Center (KAIMRC), King Abdulaziz Medical City, National Guard Health Affairs, Riyadh 11426, Saudi Arabia; 5Nanomaterials for Bioimaging Group (nanoBIG), Departamento de Física de Materiales, Facultad de Ciencias, Universidad Autónoma de Madrid (UAM), 28049 Madrid, Spain

**Keywords:** magnetic hyperthermia, iron oxide nanoparticles, magnetite, SAR, ILP

## Abstract

Magnetite nanoparticles (MNPs) exhibit favorable heating responses under magnetic excitation, which makes them particularly suited for various hyperthermia applications. Herein, we report the detailed self-heating mechanisms of MNPs prepared via the *Ko-precipitation Hydrolytic Basic* (KHB) methodology. The as-prepared MNPs were fully characterized using various spectroscopic techniques including transmission electron microscopy (TEM), dynamic light scattering (DLS), *X*-ray diffraction (XRD), energy-dispersive *X*-ray spectroscopy (EDX), and vibrating sample magnetometry (VSM). MNPs exhibited stable 15 nm quasi-spherical small-sized particles, pure crystalline cubic Fe_3_O_4_ phases, high saturation magnetizations (Ms = ~40 emu·g^−1^), and superparamagnetic behavior. In response to alternating magnetic fields (AMFs), these MNPs displayed excellent self-heating efficiencies with distinctive heating responses, even when minimal doses of MNPs were used. Heating efficacies and specific absorption rate (SAR) values as functions of concentration, frequency, and amplitude were systematically investigated. Remarkably, within only a few minutes, MNPs (2.5 mg/mL) showed a rapid dissipation of heat energy, giving a maximum intrinsic loss power (ILP) of 4.29 nHm^2^/kg and a SAR of 261 W/g. Hyperthermia temperatures were rapidly reached in as early as 3 min and could rise up to 80 °C. In addition, Rietveld refinement, Langevin, and linear response theory (LRT) models were studied to further assess the magnetic and heating mechanisms. The LRT model was used to determine the Néel relaxation time (*τ_R_* = 5.41 × 10^−7^ s), which was compared to the Brownian relation time value (*τ_B_* = 11 × 10^−7^ s), showing that both mechanisms are responsible for heat dissipated by the MNPs. Finally, the cytotoxicity assay was conducted on aqueous dispersions of MNPs, indicating their biocompatibility and low toxicity. Our results strongly suggest that the as-prepared Fe_3_O_4_ MNPs are promising vehicles for potential magnetically triggered biomedical hyperthermia applications.

## 1. Introduction

Magnetic hyperthermia, also known as magnetic fluid hyperthermia (MFH), requires the use of heat dissipated by magnetic nanoparticles under an alternating magnetic field (AMF) to selectively damage and kill cancer cells [1]. Generally, cancer cell death is induced in tumor cells at temperatures ~42–43 °C, while abnormal and healthy cells alike suffer irreversible damage at temperatures above 46 °C [2]. In MFH, the main challenges lie in obtaining MNPs of specific characteristics (i.e., high heating efficiencies and large specific absorption rate (SAR) values), as well as the optimization of AMF exposure under various experimental conditions. Previous reports have shown that SAR values are influenced by saturation magnetization (M_s_), particle sizes, crystallinity, MNP concentrations, as well as the amplitude and the frequency of the applied magnetic field [3,4,5,6,7]. It was specifically found that the SAR is affected by parameters such as magnetic interactions between particles, which influence the magnetic moment rotation that is responsible for heat dissipated through Brownian and Néel relaxation mechanisms [8]. The frequency and field amplitude of the AMF also have direct effects on the heating efficiencies of the MNPs due to the relaxation of their rotating magnetic moments.

On the other hand, and for an efficient clinical utilization of MNPs in hyperthermia, the heat dissipation should be optimized by using a minimal dosage of the MNPs (to ensure biocompatibility) with high saturation magnetization (to guarantee efficient heating). In addition, the chosen frequency and field amplitude should satisfy medical safety conditions (H_0_ × *f* ≤ 5 × 10^9^ A/m·s) [9,10]. Therefore, designing tailored MNPs that can dissipate heat at low concentrations under different ranges of frequencies and magnetic fields is essential to achieve controllable and efficient hyperthermia treatments. Taking these facts into consideration, we herein evaluated the self-heating abilities of Fe_3_O_4_ MNPs synthesized by our *Ko-precipitation Hydrolytic Basic* (KHB) methodology to be utilized for efficient MFH. We systematically investigated the effects of concentration, field amplitude, and frequency on SAR values of Fe_3_O_4_ MNPs. In addition, a theoretical physical background for hyperthermia analysis is briefly reviewed. Finally, the plausible heating mechanism responsible for the generation of heat from the obtained NPs is addressed by using the linear response theory (LRT) model based on the experimental results of the SAR.

## 2. Materials and Methods

### 2.1. Materials

Unless otherwise indicated, all chemicals and solvents were obtained from commercial suppliers and used as supplied without further purification. Iron (III) chloride hexahydrate (FeCl_3_·6H_2_O), iron (II) chloride tetrahydrate (FeCl_2_·4H_2_O), hexylamine, and 28% ammonium hydroxide (NH_4_OH) were all purchased from the supplier UFC Biotechnology.

Phosphate-buffered saline (PBS), advanced Dulbecco’s modified eagle medium (DMEM), phenol-red free DMEM, fetal bovine serum (FBS), and penicillin–streptomycin (Pen-Strep) were all purchased from UFC Biotechnology (Buffalo, NY, USA). Thiazolyl blue tetrazolium bromide (MTT) powder was purchased from Bioworld, USA. Cell lines were purchased from the American Type Culture Collection (ATCC) and grown in DMEM supplemented with 10% FBS and 1% penicillin–streptomycin. The human cancerous cells used in this study were derived from MDA-MB-231 (the metastatic breast cancer cell line isolated at MD Anderson from a pleural effusion of a 51-year-old Caucasian woman with invasive ductal carcinoma).

### 2.2. Preparation of Bare-Fe_3_O_4_ MNPs by KHB Method [10]

FeCl_3_·6H_2_O (0.250 g, 0.925 mmol) was dissolved in water (5 mL) and stirred for a few minutes at 80 °C under argon conditions. To this solution, aqueous FeCl_2_·4H_2_O (0.100 g, 0.500 mmol) was injected followed by the slow addition of ~50 μL hexylamine. Ammonium hydroxide NH_4_OH 28% (~1.5 mL) was finally added where the formation of black Fe_3_O_4_ MNPs was evident. Stirring was continued for additional 3 h. The NP suspensions were then purified via centrifugation (4500 rpm, 10 min); washed several times with isopropanol, ethanol, and water; and finally re-dispersed in water to form stable aqueous dispersions of Fe_3_O_4_ MNPs (10 mg/mL).

### 2.3. Characterization

XRD analysis was performed using the Bruker D8 Discover diffractometer (θ-2θ), equipped with Cu-Kα radiation (λ = 1.5406 Å). The morphology of the samples was studied by means of TEM (Type JEOL JSM-200F atomic resolution microscopy operating at 200 kV, Tokyo, Japan). High-resolution TEM (HR-TEM, Tokyo, Japan) was recorded on a JEOL-2100 electron microscope operating at an accelerating voltage of 200 kV. The different elements and their percentages in the sample were determined by an energy-dispersive *X*-ray spectrometer (EDX, Tokyo, Japan). Magnetic characterization was performed using a vibrating sample magnetometer (VSM, 7404 model, LakShore Westerville, OH, USA) with 1.8 T magnets at an ambient temperature.

### 2.4. Self-Heating

The heating efficiency of the samples was performed using a commercial system “Nanotherics Magnetherm”, as reported in our previous work [11]. The influence of the concentration of MNPs was investigated at 170 Oe and 332.8 kHz for the field amplitude and frequency, respectively. A selected concentration of 5 mg/mL was used to study the influence of the field amplitude (60–170 Oe) and frequency (113, 332, and 630 kHz) on the heating ability of the MNPs. The samples were dissolved in distilled water and sonicated for 5 min and the temperature increase in the samples was then recorded for 15 min. The heat generated by MNPs under an AMF was quantified by the SAR, which can be determined by:
(1)$$SAR=\frac{\rho C_{w}}{Mass_{MNP}}\left(\frac{\Delta T}{\Delta t}\right)$$
where *C_w_* is the specific heat capacity of water (4.185 kJ/kg·K), *ρ* is the density of the colloid, $Mass_{MNP}$ is the concentration of the magnetic nanoparticles in suspension, and $\frac{\Delta T}{\Delta t}$ is the heating rate. By performing a linear fit of temperature increase vs. the time at the initial time interval (1 to 30 s), the slope ΔT/Δt was obtained.

### 2.5. Cell Viability Assay

The cell viability of MDA-MB-231 cells treated with MNPs was determined using the MTT assay. The cell lines were seeded in a 96-well plate at a density of 5 × 10^5^ cells/well and incubated in 95%/5% humidified air/CO_2_ at 37 °C. After 24 h, the media were removed and fresh phenol-red free DMEM containing 0.5% FBS was added to the cells. The cells were then treated with different concentrations of Fe_3_O_4_-MNPs (1, 0.5, and 0.25 mg/mL) along with a free anticancer drug, doxorubicin (5 µg/mL), to display cell death. After 24 h of incubation, the media were removed and the cells were washed with PBS. The cell viability was then determined using the MTT viability assay in accordance with the manufacturer’s protocol. Briefly, 20 µL of the MTT reagent (5 mg/mL) was added to each well and kept for 4 h at 37 °C in the incubator. The supernatant was then removed, and the MTT formazan was dissolved in dimethyl sulfoxide. The absorbance was measured on an iMark micro-plate absorbance reader at 590 nm. The percentage of viable cells was calculated as the ratio of the absorbance of the treated group, divided by the absorbance of the control group, multiplied by 100. The absorbance from the untreated control cells was set as 100% viable.

## 3. Results and Discussion

### 3.1. Preparation and Characterization of Fe_3_O_4_ MNPs

MNPs were prepared using our previously reported *KHB* method [10,11], as illustrated in Figure 1. Briefly, the sequential in situ basic hydrolytic precipitation of iron salts (Fe^3+^ and Fe^2+^) in the presence of hexylamine and ammonium hydroxide NH_4_OH base afforded stable aqueous dispersions of Fe_3_O_4_-MNPs. The obtained MNPs were characterized by various electronic and spectroscopic techniques including transmission electron microscopy (TEM), dynamic light scattering (DLS), *X*-ray diffraction (XRD), and vibrating sample magnetometry (VSM). These techniques clearly revealed the structure, morphology, and magnetization of the as-synthesized Fe_3_O_4_-MNPs. TEM and high-resolution TEM (HR-TEM) images (Figure 2a) clearly indicate the quasi-spherical morphology and highly crystalline nature of the NPs, with well-defined interfringe spacings in their crystalline lattices. Figure 2b shows the particle size distribution associated with the TEM image. As shown, the narrow size distribution confirmed the monodispersity and homogeneity of the as-synthesized MNPs with average sizes of ~15 nm. The DLS of samples dispersed in water was then recorded (Figure 2c), showing an average hydrodynamic (D_H_) size of 105 ± 2.48 nm. The average polydispersity index (PDI) was found to be 0.3, further confirming the homogenous size-distribution, uniformity, and good dispersity of the MNP samples in aqueous media.

Next, XRD patterns of the as-prepared MNPs were conducted (Figure 3a). As can be seen, the patterns indicate the presence of diffraction peaks which correspond to the magnetite (Fe_3_O_4_) phase (JCPDS: 19-062). The main peaks were located at 30.16°, 35.49°, 43.01°, 53.78°, 57.21°, and 62.73° can be indexed as (220), (311), (400), (422), (511), and (440) of the cubic structure (Fd3m space group) of Fe_3_O_4_ MNPs. No additional peaks were observed, suggesting that the synthesis method leads to the formation of a pure phase of magnetite.

This corroborates well with the TEM and HR-TEM results that clearly show the excellent crystalline lattices of the as-synthesized MNPs and indicate that each particle is a well-ordered single crystal despite their small sizes. Rietveld analysis was performed using the Match program and the refinement is given in Figure 3b. An excellent fit with Fe_3_O_4_ patterns can be observed, indicating that the synthesized phase is pure magnetite, with lattice parameter a = 8.394 Å. The crystallite size was calculated by the Williamson–Hall method using the following equation:(2)$$\beta \;\cos\theta =4\varepsilon \;\sin\theta +\frac{k\lambda }{D_{}}$$
where k is the grain shape factor, which is 0.9 for spherical shape; λ is the *X*-ray wavelength (λ = 1.5418 Å); ε is the lattice strain; and D is the crystallite size.

By plotting 4sinθ versus βcosθ and using the linear fitting, a crystallite size of 11 nm was deduced from the intercept with the *y*-axis, as shown in Figure 3c. The calculated crystallite size fell within the range of the grain size (~15 nm) that was obtained from the TEM results. To further confirm the chemical content of the MNPs, energy-dispersive *X*-ray spectroscopy (EDX) was conducted. The EDX spectrum revealed the presence of Fe and O, only indicating the absence of any other impurity (Figure 3d), confirming the purity of the phase attained by XRD measurements.

To determine the magnetic behavior of the MNPs, field-dependent magnetizations were measured at room temperature. Figure 4a depicts the hysteresis loop (M–H) of the as-synthesized MNPs at 300 K. The values of saturation (*M_s_*), remanence (*M_r_*), and coercivity (*H_c_*) were found to be equal to 40.5 emu/g, 0.48 emu/g, and 7.6 Oe, respectively. These negligible remanence values indicate the superparamagnetic nature of the as-synthesized MNPs. This behavior was corroborated by the small size deduced from TEM image and deduced from XRD measurements. To further confirm the behavior of MNPs, we fitted the experimental magnetization by using the Langevin function. The magnetization of superparamagnetic particles in an external magnetic field (*H*) can be described by the following Langevin function:(3)$$M=M_{s}\coth\left(\frac{m_{np}H}{K_{B}T}\right)-\frac{K_{B}T}{m_{np}H}$$
where $M_{s}$ is the saturation, $m_{np}$ is the nanoparticle magnetic moment, *T* is the temperature (300 K), and $K_{B}$ is the Boltzman constant. Figure 4b shows that the experimental magnetization (*M*–*H*) curve fitted well with the Langevin equation, confirming the superparamagnetic behavior.

The saturation of MNPs (40.5 emu/g) was largely lower than the bulk Fe_3_O_4_ value (92 emu/g), but comparable to values reported previously for magnetite NPs [12,13]. The reduced magnetization of the MNPs compared to the bulk value could be explained by the presence of other iron oxide phases, such as maghemite (γ-Fe_2_O_3_) or hematite (α-Fe_2_O_3_), but XRD results and Rietveld analysis confirm the purity of the phase. It might also be due to the high surface-to-volume ratio, which induces surface effects such as the formation of magnetic dead layers on the surface of nanoparticles. To roughly determine the thickness of the dead layers (t), we used the following expression [14]:(4)$$M_{s}=M_{b}\left(1-\frac{6t}{D}\right)\;$$
where *M_s_* is the saturation, *M_b_* is the bulk magnetization, and D is the diameter of nanoparticles deduced from TEM measurements. It was found that the thickness of the dead layer is around 1.5 nm for our MNPs, which represents around 9% of the diameter of the MNPs.

Furthermore, the effective anisotropy constant (K_eff_) was calculated, as it is among the chief parameters which might affect the heating ability of MNPs. We used the approach to saturation law (LAS) to determine K_eff_, applying the following equation to the experimental magnetization, as shown in Figure 4c [12,15]:(5)$$M\left(H\right)=M_{s}\left(1-\frac{b}{H^{2}}\right)\;$$
where b is a parameter which is deduced from the fitting of experimental magnetization with Equation (5). K_eff_ was then determined by [16]:(6)$$K_{\mathrm{eff}}=\mu_{0}M_{s}\sqrt{\frac{15b}{4}}$$

The calculated value of K_eff_ was found to be equal to 7.485 × 10^4^ (erg/m^3^), which agrees with previously reported values for MNPs [11,17]. As shown in Equation (8) below, the anisotropy has an essential role in enhancing the Néel relaxation time; hence, higher anisotropy values lead to a higher relaxation time as well as the ability of the particles to be tuned under lower frequencies [18,19,20,21].

### 3.2. Magnetic Hyperthermia Measurements

#### 3.2.1. Effect of MNP Concentrations

The heating efficiencies of Fe_3_O_4_-MNPs under an alternating current (AC) magnetic field were then evaluated. Figure 5 shows the temperature rise in the MNPs dispersed in deionized water at different concentrations under the AMF with a frequency and amplitude of 332.8 kHz and 170 Oe, respectively. The main parameters obtained from the temperature rise are summarized in Table 1. As shown in Figure 5a, MNPs show high heating abilities and reach magnetic hyperthermia temperatures (42 °C) in relatively short times for all the concentrations. As shown in Table 1, the maximum temperature increased from 53.16 to 53.77 to 80.79 °C when the concentration increased from 2.5 to 5 to 10 mg/mL, respectively. Within 15 min, the solution with the highest concentration (10 mg/mL) reached noticeably higher temperatures, in comparison with the 2.5 mg/mL and 5 mg/mL solutions. For the 10 mg/mL MNP sample, the hyperthermia temperature (42 °C) was reached in just 2.7 min, while the concentration of 5 mg/mL took ~5 min to reach the same temperature (Table 1). Interestingly, for lower concentrations of 2.5 mg/mL, MNPs indicate a very good temperature rise and reached magnetic hyperthermia in relatively short times (~6 min). It is important to highlight that for an efficient MFH treatment, minimal doses of MNPs should be used to reach hyperthermia temperatures in short time periods. Thus, synthesizing stable well-dispersed MNPs with high heating efficiencies and SAR values in large quantities in an easy, robust, cheap, and reproducible process is highly demanded in the field. The variation in calculated SAR values with different concentrations is shown in Figure 5b. The SAR values were found to decrease with concentration as they were found to be equal to 84, 163, and 261 W·g^−1^ for 10, 5, and 2.5 mg/mL, respectively (Figure 5b). These relatively high values indicate the good heating capabilities of the prepared MNPs. Figure 5b shows that the SAR increased with the decreasing concentration of MNPs. Such behavior can be explained by the fact that there are many factors affecting the SAR values, including the saturation, core size, hydrodynamic size, viscosity, and polydispersity of MNPs in the media. Important parameters such as the size, dispersity, and surface properties of the MNPs can considerably affect the SAR [10,22]. As observed by us [10], SAR values were found to be dependent on the core sizes, with greater sizes resulting in higher SAR values. These results are in agreement with an elegant study conducted by de la Presa et al. [4], showing that there is a critical size of around 12 nm for obtaining the most effective heating. It seems there is a sweet optimal size (average core size ~10–15 nm and DH ~100–150 nm) at which heat production is at its maximum. Above this optimal size, the heating efficiency started to diminish with increases in the hydrodynamic size, viscosity, and polydispersity of the MNP colloid. This could be attributed to the increase in magnetic interparticle dipolar interaction, which leads to a reduction in Neel–Brownian relaxations. The NPs will not be able to freely rotate under the applied magnetic field and this will reduce the heating. Therefore, it can be clearly concluded that using higher concentrations is not always the best approach to obtain high SAR values for MNPs. There is always an optimal concentration where SAR values can be maximized. This is directly related to several other factors that will be discussed in the subsequent sections. Along with SAR values, the maximum obtained temperature and the time to reach the hyperthermia temperature (42 °C) are also other important parameters that assess the self-heating abilities of MNPs [21]. Nonetheless, all the used concentrations (2.5–10 mg/mL) caused relatively high SAR values and reached hyperthermia temperatures (42 °C) in relatively short times, which indicate the excellent heating efficiencies of the MNPs.

#### 3.2.2. Effect of Applied Magnetic Field Amplitude and Frequency

The effects of both the field amplitude and frequency on the self-heating ability of MNPs were investigated using the middle concentration of 5 mg/mL. Different combinations of magnetic fields and frequencies were applied. The frequency was varied to 113, 332, and 630 kHz. At each of these frequencies, the magnetic field was also varied, as shown in Figure 6a–c. In Figure 6a, the frequency was fixed at 113 kHz and the magnetic fields were varied to 100, 120, and 140 Oe. The resultant temperature–time graphs revealed a noticeable increase in the temperature slope as the magnetic field increased. For each frequency, the maximum temperature reached by the MNPs also increased with the field amplitude. For instance, at the combination (332 kHz, 140 Oe), the temperature reached 42 °C within 10 min. Applying higher magnetic field strengths (i.e., 170 Oe) caused even shorter times to reach the hyperthermia temperature (6 min). We also investigated the effect of magnetic field strength at a higher frequency of 630 kHz. As shown in Figure 6c, changing the magnetic field from 60 Oe to 120 Oe caused significant differences in self-heating behaviors. At 60 Oe, no considerable change was noticed in temperature, while raising the magnetic field to 120 Oe caused an obvious increase in the temperature slope reaching 42 °C within only 7 min. This also resulted in a higher calculated SAR value of 110 W/g (Figure 6e).

Overall, for all applied frequencies, the maximum temperature reach by the MNPs increased with field amplitude, leading to higher SAR values. As shown previously in Equation (1), the calculation of SAR values depended on the initial slope of the temperature rise. The results also show that different combinations of frequency and magnetic field lead to different self-heating behaviors. In our case, the maximum heating parameters and the highest SAR values were achieved within a combination of a 332.8 kHz frequency and a 170 Oe amplitude for a concentration of 2.5 mg/mL.

In another experiment, the magnetic field amplitude was fixed at 120 Oe and the frequency was altered to 113, 332, and 630 kHz. The same trend of temperature rises and SARs is depicted in Figure 7. For instance, at a field amplitude of 120 Oe, the SAR changed from 23 W/g at a frequency of 113 kHz to 110 W/g at a frequency of 630 kHz.

Finally, the intrinsic loss power (*ILP*), used to compare the heating efficiencies of different MNPs, was calculated by using the obtained *SAR* values and applying the following equation:(7)$$ILP=SAR/fH_{0}^{2}$$
where *f* is the frequency and H_0_ is the coercivity.

The *ILP* values for different concentrations of MNPs under different sets of experimental conditions (i.e., various concentrations, applied magnetic field strengths, and frequencies) are summarized in Table 1, Table 2 and Table 3. These values are in the range reported for commercial ferrofluid (0.15 nHm^2^/kg) [23], maghemite (1.78 nHm^2^/kg) [4,11], and magnetite (9.4 nHm^2^/kg) [22]. Remarkably, the minimal dose of MNPs (2.5 mg/mL) shows the highest ILP (4.29 nHm^2^/kg). All these results indicate that the heat dissipated by MNPs can be easily tuned by changing the concentration, field amplitude, and frequency of the AMF, as reported by many other systems [4,5,8,11]. This tuning mainly depends on the magnetic properties, particle size, crystallinity, interparticle interactions, and dispersing medium; hence, the overall synthetic methodology utilized to prepare stable high-quality aqueous dispersions of MNPs is a key factor.

#### 3.2.3. Mechanism of Heating and the LRT Model

In general, superparamagnetic MNPs emit heat under the AMF as a result of two main mechanisms, namely Néel and Brownian relaxation. In Néel relaxation, the magnetic moments are reoriented under an externally applied AMF within the particles against an energy barrier with relaxation time $\;\tau_{N}$:(8)$$\tau_{N}=\tau_{0}e^{K_{eff}V/k_{B}T}$$
where *K_eff_* is isotropy coefficient, *V* is the volume of the NPs, k_B_ is the Boltzmann constant, and $\tau_{0}$ is the time constant. In Brownian relaxation, the entire particle rotates inside the carrier fluid and the friction between particles and the fluid due to the viscosity induces heat. In this case, the relaxation time $\tau_{B}$ is given by:(9)$$\tau_{B}=\frac{3\eta V_{h}}{k_{B}T}$$
where *ƞ* is the fluid viscosity, *V_H_* is the hydrodynamic volume of the particle, *k_B_* is the Boltzmann constant, and *T* is the particle temperature.

As can be seen from the above two equations, the two mechanisms depend on intrinsic properties, such as the particle size, magnetic anisotropy, and extrinsic property (media type and viscosity).

In practice, both mechanisms may occur simultaneously and contribute to nanoparticle heating with an effective relaxation time $\tau_{R}$:(10)$$\tau_{R}=\frac{\tau_{N.}\tau_{B}}{\tau_{N}+\tau_{B}}$$

As this equation illustrates, the heating mechanism with the shorter relaxation time will dominate. Thus, Néel relaxation will dominate for smaller particles, whereas Brownian motion relaxation dominates at larger particle sizes [4,5,8].

These mechanisms have been investigated in the literature through several models. For instance, many models have been used to describe the magnetic relaxation mechanisms, calculate the time-dependent magnetization as a function of the applied magnetic field, and determine the energy absorbed by the MNPs under the AMF which depicts the SAR [24]. Among these models, the linear response theory (LRT), in which the magnetic response is linear with the applied magnetic field, is commonly employed [4,25]. This model is only satisfied when the amplitude of the applied magnetic field (H_0_) ≤ the anisotropy field of the MNPs (H_K_).

Considering the validity of the LRT model, the SAR can be calculated by using the following equations:(11)$$\;\;\;\;\;\;\;\;\;\;\;\;\;\;\;\;\;\;SAR=\frac{\mu_{0}^{2}\;M_{s}^{2}\;VH_{0}^{2}}{3\;K_{B}T\tau_{R}\rho_{MNPs}}\frac{{\left(2\pi f\tau_{R}\right)}^{2}}{1+{\left(2\pi f\tau_{R}\right)}^{2}}\;$$
(12)$$\;SAR=\alpha \frac{\tau_{R}{\left(2\pi f\right)}^{2}}{1+{\left(2\pi f\tau_{R}\right)}^{2}}\;$$
where $\alpha =\frac{\mu_{0}^{2}\;M_{s}^{2}\;VH_{0}^{2}}{3\;K_{B}T\rho_{MNPs}}$, μ_0_ is the vacuum permeability, *M_s_* is the saturation magnetization, V is the particle volume, *H*_0_ is the field amplitude, *k_B_* is the Boltzmann constant, *T* is the absolute temperature, $\rho_{MNPs}$ is the density, $\tau_{R}\;$ is the relaxation time, and f  is the frequency of the filed.

Thus, to further inspect the heat dissipation mechanism of our MNPs, we applied the LRT model to our experimental data. The LRT model is valid when the field amplitude (*H*_0_) is less than or comparable to the anisotropy field *H_K_* (*H*_0_ ≤ *H_K_*). The experimental field amplitudes used in our measurements are less or comparable to the value of *H_K_* = 10 KA/m reported for MNPs with moderate anisotropy such us magnetite [24], which justifies the validity of the LRT model. As expected by the LRT model, the experimental SAR values vary linearly as a function of the square of the field amplitude. By fitting the SAR values by *SAR =*
$\;cfH_{0}^{2}$, we observed that that the field amplitude dependence on experimental SARs displays quadratic behavior (Figure 8a). The coefficient of determination, *R*^2^, which should be near 1, is in the range 0.98–0.998 for the different frequencies, indicating the validity of the LRT model.

Furthermore, we determined the relaxation time, $\tau_{R}$, by fitting the frequency dependence of SARs using Equation (12). As depicted in Figure 8b, the SAR values fitted well with Equation (12), with a determination coefficient of *R*^2^ = 0.986. The relaxation time deduced from the fit was around $\tau_{R}=5.41\times 10^{-7}\;$s. This relaxation was close to the time reported for iron oxide NPs in the frequency range investigated here [4,26,27,28]. Interestingly, the obtained value of *τ_R_* indicates that the MNPs were in the regime 2π*fτ_R_* < 1 for all frequencies used in this work, confirming the legitimacy of LTR model. The Brownian relaxation time, τ_B_, deduced from Equation (9) was around *τ_B_* = 11.1 × 10^−7^ s. Both relaxations were in the same order, allowing us to conclude that both relaxation mechanisms affect the heat dissipated by the MNPs.

### 3.3. Cytotoxicity and Biocompatibility of MNPs

It is crucial to evaluate the cytotoxicity of as-prepared MNPs before employing them for magnetic hyperthermia applications. To this end, the toxicities of three different concentrations of MNPs towards a metastatic breast cancer cell line MDA-MB-231 were evaluated using the thiazolyl blue tetrazolium bromide (MTT) viability assay. The MTT assay is based on the capacity of the mitochondrial enzyme of viable cells to transform the MTT tetrazolium salt into a violet bluish-colored MTT formazan, which is proportional to the number of living cells present. As shown in Figure 9, Fe_3_O_4_ MNPs were not toxic to MDA-MB-231 cancerous cells, even at the high concentrations used. Moreover, 80% of the cells were still found to be viable when treated with a very high MNP concentration (up to 1 mg/mL). Nonetheless, the anticancer drug, doxorubicin, was found to be highly potent to MDA-MB-231 cancer cells, killing almost all cancer cells. This is in accordance with earlier reported evidence, by us and others, where even using 1 mg/mL of Fe_3_O_4_ MNPs was considerably safe to the cells with no significant cytotoxicity. For effective cancer hyperthermia, it is necessary that the MNPs produce a maximum temperature rise at the lowest concentration possible in relatively short times and clinically relevant AC magnetic fields (Hergt’s limit H_0_ *f* < 5 × 10^9^ A/m^−1^·s^−1^). Thus, the concentrations used here (~1 to 10 mg/mL NP—significantly lower than the concentrations of other superparamagnetic NPs used in the literature), along with the high heating efficiencies and SARs, strongly suggest the promising potential of using prepared Fe_3_O_4_-MNPs for magnetic hyperthermia applications.

## 4. Conclusions

In conclusion, a simple, cheap, and reproducible KHB methodology was used to synthesize Fe_3_O_4_ NPs with noticeable aqueous stability, small sizes, superparamagnetic behaviors, and excellent heating efficiencies. The heating ability of the MNPs was investigated as a function of the concentration of MNPs, field amplitude, and frequency. It was found that MNPs reach hyperthermia temperatures (42 °C) in relatively short times for all the tested concentrations. Interestingly, for low concentrations of 2.5 mg/mL, the highest SAR (261 W/g) and ILP (4.29 nHm^2^/kg) values were obtained, where magnetic hyperthermia temperatures were reached in a short time (~6 min). Furthermore, it was found that SAR values have a quadratic dependence on the field amplitude, validated by the LRT model. A value of 5.41 × 10^−7^ was obtained for relaxation times *τ_N_* by fitting the SARs as a function of the field frequency. This value is comparable to that calculated for Brownian relaxation times *τ_B_* (11 $\times \;10^{-7}$). Finally, the viability of the as-synthesized MNPs was tested against metastatic breast cancer cells, showing minimal toxicity, even at the relatively high concentrations employed. The high crystallinity, high ILP (4.29 nHm^2^/kg), good SAR (261 W/g), and low toxicities obtained here strongly suggest that the as-prepared magnetite NPs are promising candidates for biomedical magnetic hyperthermia applications.

## Figures and Tables

**Figure 1 nanomaterials-13-00453-f001:**
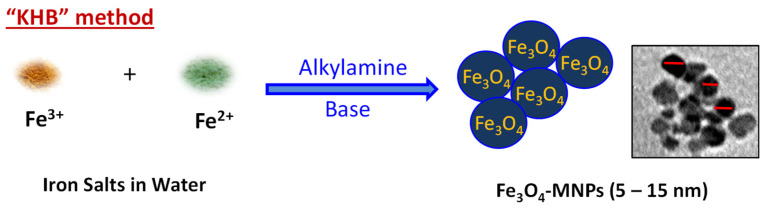
Schematic representation for the synthesis of iron oxide magnetic nanoparticles (Fe_3_O_4_-MNPs) using Ko-precipitation Hydrolytic Basic (KHB) methodology.

**Figure 2 nanomaterials-13-00453-f002:**
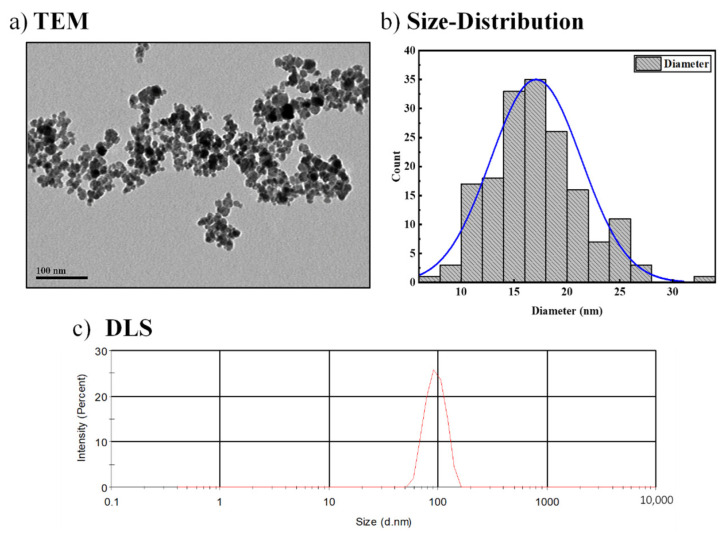
(**a**) TEM image; (**b**) corresponding particle size distribution; and (**c**) hydrodynamic size (D_H_) distribution of Fe_3_O_4_-MNP dispersed in water. TEM and HR-TEM images clearly show the highly crystalline nature of the as-synthesized MNPs and their well-defined interfringe spacings.

**Figure 3 nanomaterials-13-00453-f003:**
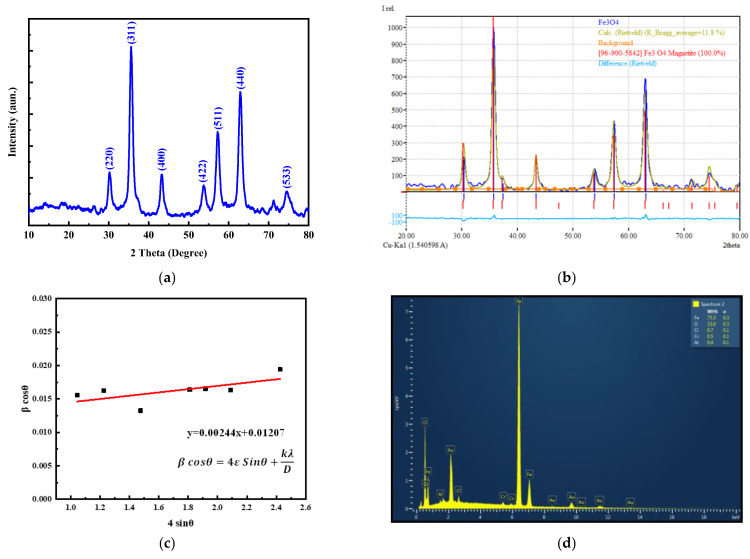
(**a**) XRD patterns; (**b**) Rietveld analysis; (**c**) Williamson–Hall method; and (**d**) EDX spectrum of the synthesized Fe_3_O_4_ MNPs.

**Figure 4 nanomaterials-13-00453-f004:**
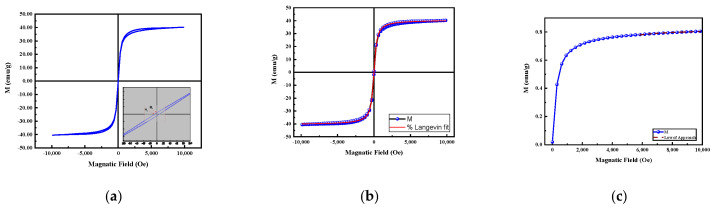
(**a**) Hysteresis loop at room temperature; (**b**) fitting of the experimental magnetization with the Langevin function; and (**c**) the saturation law.

**Figure 5 nanomaterials-13-00453-f005:**
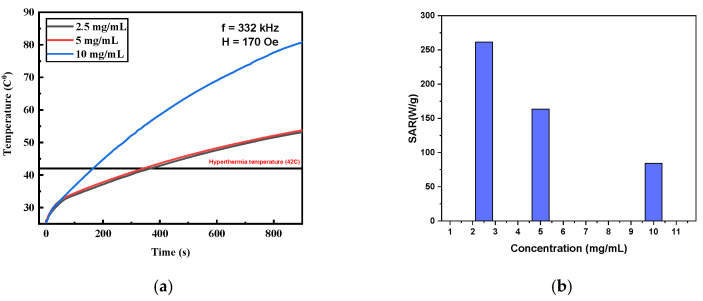
(**a**) Temperature rise for different concentrations and (**b**) SAR values as function of concentration.

**Figure 6 nanomaterials-13-00453-f006:**
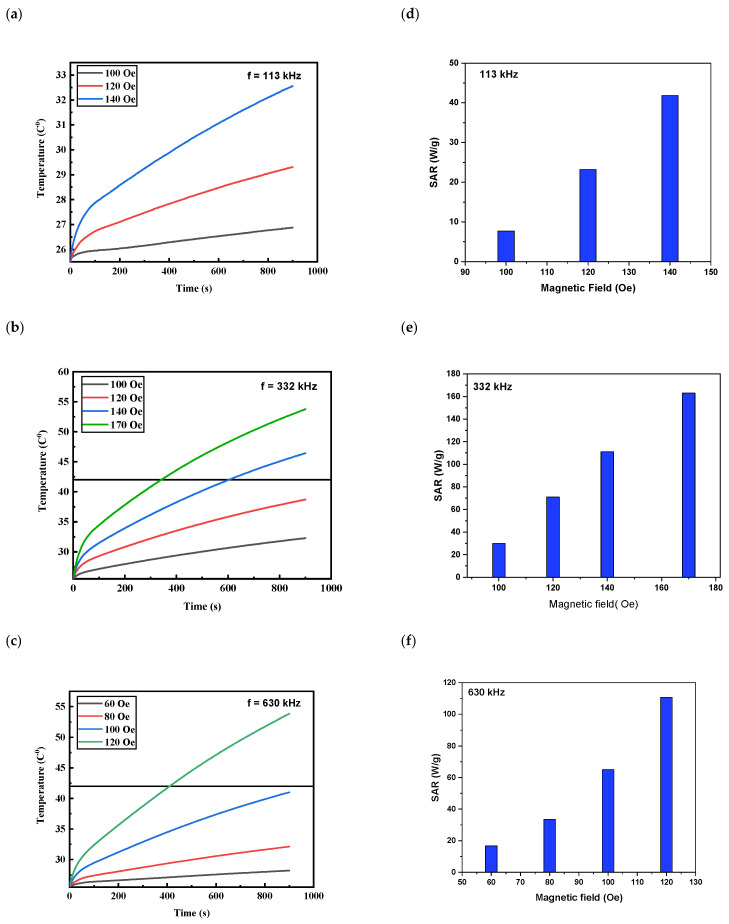
(**a**) Temperature increases at different frequencies and field amplitudes for a concentration of 5 mg/mL (**a**–**c**) and (**d**–**f**) SAR values.

**Figure 7 nanomaterials-13-00453-f007:**
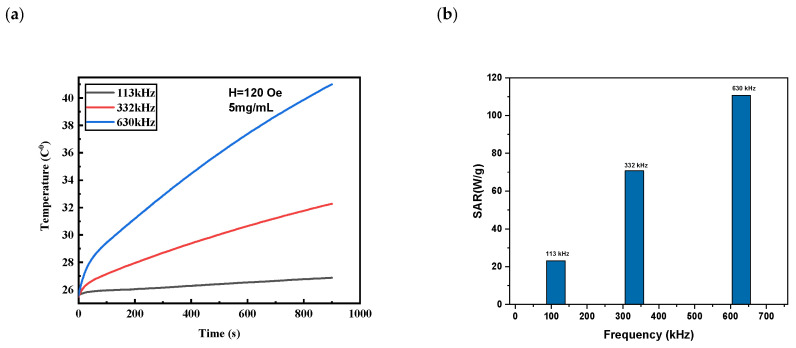
(**a**) Temperature rise at different frequency and field amplitude of 120 Oe for concentration of 5 mg/mL and (**b**) SAR values.

**Figure 8 nanomaterials-13-00453-f008:**
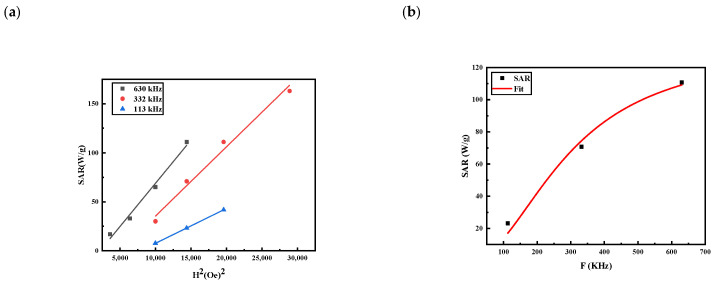
LRT model fitting the experimental SARs (**a**) with the square of field amplitude and (**b**) with the frequency at field *H*_0_ = 120 Oe using Equation (12).

**Figure 9 nanomaterials-13-00453-f009:**
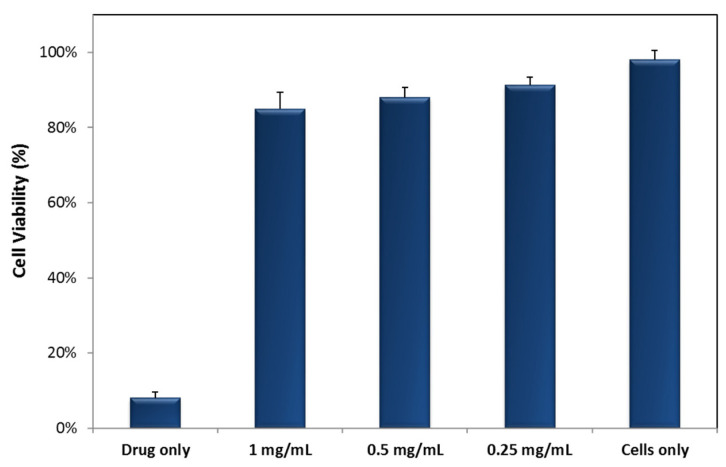
MTT cell viability assay for MDA-MB-231 metastatic breast cancer cells incubated with different concentrations of Fe_3_O_4_ MNPs or the free anticancer drug doxorubicin. The results clearly show the low toxicity of MNPs, while the free drug is shown to be considerably toxic, killing almost all the cells.

**Table 1 nanomaterials-13-00453-t001:** Heating characteristics for different concentrations of Fe_3_O_4_ MNPs at a field amplitude *H*_0_ = 170 Oe and a frequency *f* = 332.8 kHz.

Concentration (mg/mL)	Maximum Temperature (°C)	Time Needed to Reach Hyperthermia Temperature 42 °C (min)	SAR (W/g)	ILP
2.5	53.16	6.12	261.21	4.29
5	53.77	5.70	163.42	2.68
10	80.79	2.77	84.28	1.38

**Table 2 nanomaterials-13-00453-t002:** Heating characteristics for Fe_3_O_4_ MNPs (5 mg/mL) at different field amplitudes and *f* = 332.8 kHz.

Field *H*_0_ (Oe)	Maximum Temperature (°C)	Time Needed to Reach Hyperthermia Temperature 42 °C (min)	SAR (W/g)	ILP
100	32.28	Not reached	30.24	1.43
120	38.71	Not reached	70.77	2.33
140	46.43	10.05	111.31	2.69

**Table 3 nanomaterials-13-00453-t003:** Heating characteristics for Fe_3_O_4_ MNPs (5 mg/mL) at different frequencies and field amplitudes *H*_0_ = 120 Oe.

Frquency (kHz)	Maximum temperature (°C)	Time Needed to Reach Hyperthermia Temperature 42 °C (min)	SAR (W/g)	ILP
113	29.31	Not reached	23.16	2.93
332	38.71	Not reached	70.77	2.33
630	53.84	6.83	110.66	1.94

## Data Availability

The data presented in this study are available within the article.

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
