# Peer review of "Assessing the Heat Generation and Self-Heating Mechanism of Superparamagnetic Fe3O4 Nanoparticles for Magnetic Hyperthermia Application: The Effects of Concentration, Frequency, and Magnetic Field"

_nanomaterials, 2023, doi:10.3390/nano13030453_

Round 1

Reviewer 1 Report

The manuscript entitled, ‘Assessing the heat-generation and self-heating mechanism of superparamagnetic Fe3O4 nanoparticles for magnetic hyperthermia application: effects of concentration, frequency, and magnetic field’ discussed the effect of magnetic nanoparticles in hyperthermia based applications. I am mentioning some points which should be justified before publication;

1.      Is this effect is applicable only for magnetite types or all types of MNPs?

2.      What will be effect for superparamagnetic materials?

3.      SAR is dependent over the alternating magnetic field frequency and domain size. Did the author consider both or one? Argument is needed.

4. Some articles have significance with this report like; https://doi.org/10.1016/j.progpolymsci.2022.101574; https://doi.org/10.1002/pat.5344; https://doi.org/10.1016/j.ijthermalsci.2022.107887; https://doi.org/10.1115/1.4056487.

Reviewer 2 Report

The manuscript by Lemine and coworkers describes the preparation and characterization of superparamagnetic iron oxide nanoparticles (MNPs) and an assessment of their suitability for magnetic hyperthermia applications.  The detailed characterization of the MNP properties is a strength of the paper, although there are a few areas where additional details and explanation are required.  The main problem with the paper is that it does not provide much perspective on the advantages of these MNPs compared to other similar materials.  The conclusions need to provide more perspective on this question.  Which (if any) of the studied properties indicate improved suitability for magnetic hyperthermia applications? 
Other specific comments/questions are summarized below.

Abstract
The first sentence should be rewritten.  Perhaps something like:  “Magnetite nanoparticles (MNPs) exhibit favorable heating responses under magnetic excitation, which makes them particularly well-suited for various hyperthermia applications.” 
Abbreviations used in the Abstract should be defined. 
Line 47.  A single cytotoxicity assay does not “prove” biocompatibility—perhaps “indicates” or “suggests” is more appropriate. 
Line 49.  It is not clear what is meant by combined magnetically-triggered biomedical hyperthermia applications—combined with what?

Line 80.  What is the meaning of Ko in the KHB method?

Section 3.1.  The individual MNPs are not very clear in the TEM image shown; possibly an image or an inset at higher magnification can be shown.  Furthermore, the particles appear strongly aggregated, making it hard to distinguish between individual particles.  The experimental section should provide some details on exactly how the “diameter” is determined and what particles are analyzed. It would also be useful to provide the mean diameter and standard deviation for panel b.    
Lines 164-165 state that the TEM and DLS confirm a homogeneous size distribution, uniformity and good dispersity of the MNPs.  That is not especially evident in the data shown.  The particles are certainly aggregated in the TEM image and based on the large hydrodynamic diameter measured by DLS, they are also aggregated in aqueous media.  For the DLS data, the polydispersity should be provided as a measure of the size distribution. 
Figure 2.  The quality of panel b (Rietveld analysis) is very poor and should be improved.  It is also challenging to read the labels on the EDX plot in panel d.  The agreement between the TEM and XRD particle size is actually impressive which could be emphasized. 

Section 3.2.2.  The text in lines 299-300 notes that the ILP values (Table 1-3) are comparable to those reported for other materials.  It would be useful to provide typical values for these materials so that the reader can assess the similarity. 

Section 3.2.3.  The last 2 paragraphs refer to Figure 7, but I think this should be Figure 8.  An equation number is missing from the caption for Figure 8. 

Section 3.3.  The biocompatibility has been assessed with a single assay.  However, it seems a missed opportunity to not report preliminary results using the MNPs to demonstrate that they are suitable for magnetic hyperthermia. 
Line 374.  Figure 9, not Figure 8.  The figure caption should indicate what drug only means and the drug concentration should be given either in the caption or the Experimental section. 

Reviewer 3 Report

The paper by Lemine et al. brings extensive characterization of magnetic nanoparticles for hyperthermia applications and is very suitable for Nanomaterials journal.  Some minor changes are desirable:

Line 45: relation -> relaxation;

Line 129: Equation numbering should be right justified;

Line 130: J/g.k -> J/goC;

Line 177: Sin -> sin;

Line 189: Symbols for all physical quantities should be written in italics;

Lines 339 and 340: This is one equation and (12) should be deleted;

Lines 397 and 399: Insert s at appropriate places.

My main concern is about "the temperature of the MNP dispersed in deionized water" (line 233). The temperature actually measured is that of the dispersion, i.e. of the water and not of the MNPs themselves. Each nanoparticle under the described conditions is a heater and its temperature is relevant quantity for the proposed applications. After some time an equilibrium should be established whereby the temperature of the nanoparticles and the water are the same. This equilibrium is not seen in the figures. The quantity of the released heat into the medium should be proportional to the number of heaters, i.e. nanoparticles. However, in Table 1 it is seen that "the maximum temperature" is not proportional to concentration.

The authors should add an additional piece of text clarifying this issue.

Round 2

Reviewer 1 Report

This can be published in its present form.

Reviewer 3 Report

The manuscript has been improved and certain points clarified. It is now possible to recommend it for publishing.